# Beyond Sequential Context: Navigating Non-linear Flow of Multi-turn Dialogues with Dynamic Context Tree

## Abstract

Large Language Models demonstrate outstanding performance in many language tasks but still face fundamental challenges in managing the non-linear flow of human conversation. The prevalent approach of treating dialogue history as a flat, linear sequence is misaligned with the intrinsically hierarchical and branching structure of natural discourse, leading to inefficient context utilization and a loss of coherence during extended interactions involving topic shifts or instruction refinements. To address this limitation, we introduce Context-Agent, a novel framework that models multi-turn dialogue history as a dynamic tree structure. This approach mirrors the inherent non-linearity of conversation, enabling the model to maintain and navigate multiple dialogue branches corresponding to different topics. Furthermore, to facilitate robust evaluation, we introduce the Non-linear Task Multi-turn Dialogue (NTM) benchmark, specifically designed to assess model performance in long-horizon, non-linear scenarios. Our experiments demonstrate that Context-Agent enhances task completion rates and improves token efficiency across various LLMs, underscoring the value of structured context management for complex, dynamic dialogues.

## 1 INTRODUCTION

The advancement of dialogue systems based on Large Language Models (LLMs) is pivotal for the efficacy of next-generation applications, including complex AI Agents and collaborative robotics, where the ability to maintain context-aware communication is fundamental to task completion and user engagement (Durante et al., 2024; Yao et al., 2024). Following the advent of LLMs' context window expansion techniques, the capabilities for multi-turn dialogue have been significantly enhanced (Li et al., 2025).

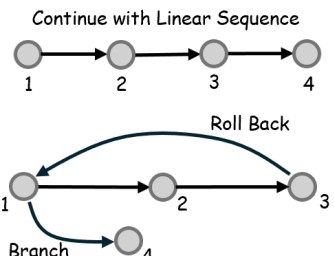

Figure 1: A schematic diagram of linear (upper) vs. non-linear (lower) dialogue flow.

However, LLMs still grapple with a fundamental challenge inherent to natural human conversation: the management of non-linear dialogue flow. This phenomenon occurs when conversational topics do not advance in a sequential order but instead feature shifts, topical jumps, or interwoven threads of discussion (Laban et al., 2025). Such non-linear dynamics are commonplace in real-world interactions, where users may revisit previous topics, introduce new subjects, or refine earlier statements based on evolving understanding or context (Mann & Thompson, 1988). The prevalent approach of treating dialogue history as a flat, linear sequence is fundamentally misaligned with the intrinsic structure of human conversation (Wang et al., 2024; Li et al., 2025). This linear paradigm fails to capture the hierarchical and branching nature of dialogues, leading to inefficiencies in context utilization and challenges in maintaining coherence over extended interactions (Ding et al., 2024).

Effectively resolving the non-linear flow problem requires overcoming several distinct challenges. The first is the accurate identification and management of topic shifts and instruction refinements within a conversation. The second challenge is the efficient selection of context from a potentially

vast and complex dialogue history. As conversations extend over multiple turns, the accumulation of information can lead to increased computational costs and the risk of overwhelming the model with irrelevant details (Joren et al., 2025), leading to the "needle in a haystack" problem (Liu et al., 2024b; Vaswani et al., 2017). The third challenge lies in the development of robust evaluation metrics and benchmarks that can accurately assess a model's performance in handling non-linear dialogues, as existing datasets often lack the complexity and variability found in real-world interactions.

To address these challenges, inspired by the hierarchical organization inherent in human cognitive processes for managing complex dialogues (Grosz & Sidner, 1986), we propose Context-Agent, a novel framework that models multi-turn dialogue history as a dynamic tree structure. This approach allows for the representation of conversations in a way that reflects their inherent non-linear nature, enabling the model to maintain multiple branches of dialogue corresponding to different topics or sub-topics. Furthermore, recognizing the inadequacy of existing datasets for this problem, we introduce the Non-linear Task Multi-turn Dialogue (NTM) benchmark, specifically designed to evaluate the performance of models in long-horizon, non-linear dialogue scenarios. This benchmark features dialogues with multiple topic shifts and instruction refinements, providing a more realistic and challenging testbed for assessing context management strategies.

In summary, the main contributions of this paper are as follows:

- We propose Context-Agent, a novel framework that models dialogue history as a dynamic tree structure and integrates it with a retrieval-augmented generation (RAG) mechanism. This enables efficient, structurally-aware context selection, allowing the model to accurately identify and provide the most relevant conversational branches for each query.

- We introduce a new benchmark, the Non-linear Task Multi-turn Dialogue (NTM), specifically designed to evaluate the performance of models in long-horizon, non-linear dialogue scenarios. This benchmark features dialogues with multiple topic shifts and instruction refinements, providing a more realistic and challenging testbed for assessing context management strategies.

- We conduct extensive experiments on the NTM benchmark, demonstrating that our Context-Agent framework outperforms mainstream context management methods across various LLMs. Notably, it achieves improvements in task completion rates while reducing token usage, highlighting its effectiveness and efficiency in managing complex non-linear multi-turn dialogues.

## 2 RELATED WORK

**Architectural Context Extension Methods**  To overcome Transformer context window limits, recent work has improved position encoding and attention mechanisms (Tworkowski et al., 2023). **Position Interpolation (PI)** (Chen et al., 2023) rescales position indices, while **YaRN** (Peng et al., 2024) extends Rotary Position Embeddings (RoPE) for longer contexts. Efficient attention methods like **LongLoRA** (Chen et al., 2024) use Shifted Sparse Attention to reduce computation. However, these approaches only enlarge the context window and do not address efficient organization or retrieval of relevant content, so issues like high cost and the "lost in the middle" problem remain.

**Information Compression and Selection Methods**  Another line of work compresses dialogue history to balance efficiency and information retention. Typical approaches combine clustering and summarization. For example, (Su & Zhou, 2022) use spectral clustering to group turns by topic and summarize each cluster. Context distillation methods, such as (Park et al., 2021), train compact models to preserve key relationships, achieving significant memory savings with minimal performance loss. However, these methods treat history as an unstructured sequence, relying only on semantic similarity and failing to capture topic shifts or hierarchical sub-topics essential for coherence in complex dialogues.

**Retrieval-Augmented Context Selection**  Recent advances adapt **Retrieval-Augmented Generation (RAG)** from external knowledge retrieval to internal dialogue history (Lewis et al., 2020). For example, **DH-RAG** (Zhang et al., 2025) builds a dynamic database of "query-passage-response" triplets and uses clustering, hierarchical matching, and Chain-of-Thought tracking for retrieval. However, its flat structure limits modeling of dialogue flow, as retrieval relies solely on semantic similarity, not structural relationships.

In summary, prior work lacks structured representations of dialogue history, often relying on unstructured or domain-specific approaches, which limits their generalizability to complex conversations. Our work addresses this gap and opens a new direction for efficient, general context management in multi-turn dialogue.

## 3 METHOD

Our framework models a multi-turn dialogue as a forest of topic trees. Each tree represents a distinct topic and is composed of nodes (dialogue units) and branches. The dialogue's evolution is managed through state transitions.

### 3.1 FORMAL PROBLEM DEFINITION

Conventional dialogue systems model history as a linear sequence $H_t = \{(q_1, r_1), \ldots, (q_t, r_t)\}$, generating a response $r_{t+1}$ from a query $q_{t+1}$ via a function $g(H_t, q_{t+1})$. This flat representation leads to contextual redundancy and loss of structural information.

To address this limitation, we introduce and formalize the problem of Non-linear Contextual Dialogue Management. The central premise of this problem is to shift from treating the entire history $H_t$ as an undifferentiated input to representing it as a dynamically evolving, hierarchically structured dialogue forest, denoted as $F_t$. At each turn $t + 1$, given:

- A structured dialogue history represented as a forest, $H_t = F_t$.
- The current dialogue state $S_t = (H_t, T_{\text{act}}, B_{\text{act}}, n_{\text{cur}})$, which includes the history, the active topic tree, the active branch, and the current node.
- The new user query $q_{t+1}$.

The objective is to learn a policy $\pi$ that comprises two key functions: a context selection function, $f_{\text{select}}$, and a response generation function, $f_{\text{gen}}$:

$$C_{t+1} = f_{select}(q_{t+1}, S_t) \qquad r_{t+1} = f_{gen}(q_{t+1}, C_{t+1})$$

Here, $C_{t+1}$ represents a highly relevant context subset, which is dynamically selected and constructed from the structured history $H_t$. The ultimate goal is to maximize the task completion rate while minimizing the token footprint of the selected context $C_{t+1}$, thereby achieving efficient context utilization without compromising conversational coherence or task-oriented performance.

### 3.2 CORE COMPONENTS

**Node** The smallest unit of a conversation is a node $n$, which represents the content of a round of dialogue between the user and the model. Each node is defined as a tuple:

$$n = (c, v, p, \beta, s_i)$$

where $c$ is the content of the current conversation round, $v \in \mathbb{R}^d$ is its $d$-dimensional text embedding, $p$ is the parent node's identifier (null for a root), $\beta$ is the branch identifier, and $s_i$ is a summary of the node's content. After each round, a summarization function $S_{node}$ converts the content $c_i$ into a summary $s_i = S_{node}(c_i)$, which is used for subsequent topic attribution and branch management.

**Topic Tree** An independent topic is represented by a topic tree $T$. It is a directed acyclic graph, $T = (N, E)$. Here, $N = \{n_1, n_2, \ldots, n_k\}$ is the set of all nodes under this topic, and $E = \{(n_i, n_j) \mid p(n_j) = n_i\}$ is the set of directed edges between nodes, representing the inheritance relationship of the conversation. The first dialogue round of a new topic is set as the root node, whose parent node is null, of the topic tree.

**Branch** Within the same topic tree $T$, a branch $B$ is a relatively independent dialogue path that starts from a branching point but still remains under the same topic. It is defined as an ordered sequence of nodes $B = \langle n_1, n_2, \ldots, n_k \rangle$, where any two adjacent nodes $(n_i, n_{i+1})$ in the sequence satisfy $p(n_{i+1}) = n_i$. All nodes within the same branch share the same branch identifier $\beta$.

**Conversation History** The complete history $H$ of a multi-turn conversation is represented as a forest $F$ consisting of multiple topic trees, i.e., $H = F = \{T_1, T_2, \ldots, T_m\}$.

### 3.3 STATE TRANSITION

The conversational state at turn $t$ is defined as $S_t = (H_t, T_{act}, B_{act}, n_{cur})$, which includes the history, the active topic tree, the active branch, and the current node. The conversation evolves through state transitions driven by new user queries. Upon receiving a new query, the system analyzes it to determine the topic and manage branches, updating the state accordingly. This process involves the following steps:

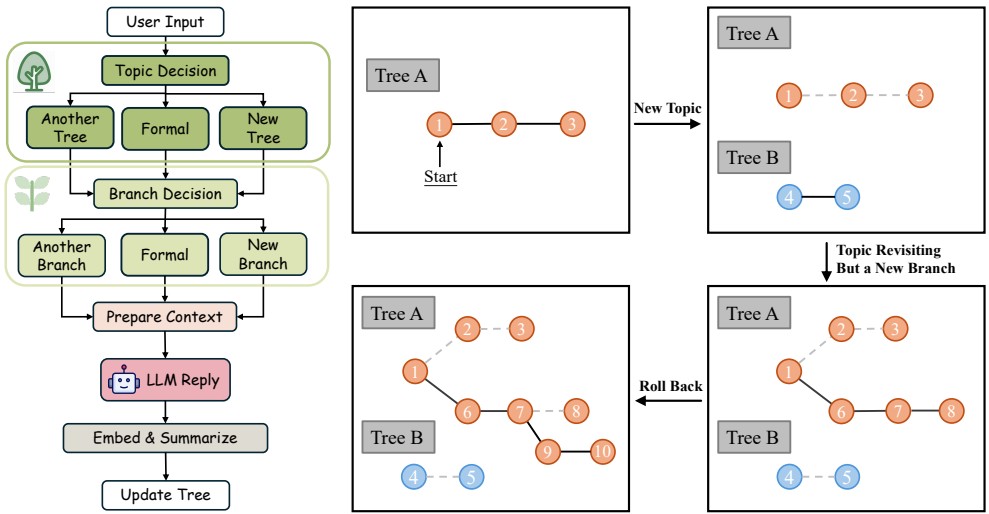

Figure 2: An overview of the Context-Agent framework. It illustrates the dynamic evolution of a multi-turn dialogue represented as a forest of topic trees, with branches indicating sub-dialogue paths. The number in each node represents the turn number in the conversation. And solid edges represent the active path, while dashed edges indicate inactive paths.

- **Step0: Initialization** The dialogue begins by creating the first topic tree $T_1$, which becomes the active tree $T_{act}$. An aggregation summary function $S$ is defined to produce summaries for branches or trees by concatenating the summaries of their constituent nodes, i.e., $S(B) = \text{Concat}(s_1, \ldots, s_k)$ for a branch $B$ and $S(T)$ for a topic tree $T$.

- **Step1: Topic Decision** For a new query $q_{t+1}$, a lightweight language model $\Psi$ determines the topic action $a_{\text{topic}}$ and target tree $T_{\text{target}}$ based on summaries of existing trees:

$$(a_{\text{topic}}, T_{\text{target}}) = \Psi(q_{t+1}, \{S(T_i)\})$$

The active tree $T_{\text{act}}$ is then updated to $T_{\text{target}}$. The action $a_{\text{topic}}$ can be:

- **CREATE_TOPIC**: A new topic tree is created.
- **SWITCH_TOPIC**: Switch to the most relevant existing tree.
- **CONTINUE**: Remain in the current tree.

- **Step2: Fork Point Identification** For a new query $q_{t+1}$, the system first computes its embedding vector $v_{q,t+1} = \epsilon(q_{t+1})$ using the embedding function $\epsilon : C \to \mathbb{R}^d$. Then, among all nodes in the active topic tree $T_{act}$, it identifies the node most semantically relevant to $q_{t+1}$ as the potential fork point. This is achieved by maximizing the similarity function $Sim(v_{q,t}, v_i)$:

$$n_{fork}^* = \arg\max_{n_i \in N_{act}} \text{Sim}(v_{q,t+1}, v_i)$$

- **Step3: Branch Decision** Branch decision employs a two-stage "heuristic filtering + model decision" approach. First, a heuristic function $H_{\text{filter}}$ quickly determines if a complex decision is needed. If the most similar node $n_{fork}^*$ found in Step 2 is sufficiently relevant (similarity $> \theta_{\text{sim}}$) and belongs to a different branch or is an ancestor, $H_{\text{filter}}$ returns true.

$$H_{\text{filter}} := (\text{sim}_{\max} > \theta_{\text{sim}}) \wedge (\beta(n_{fork}^*) \neq \beta(n_{\text{cur}}) \vee \text{IsAncestor}(n_{fork}^*, n_{\text{cur}}))$$

If $H_{\text{filter}}$ is true, a lightweight language model $\Phi$ determines the branch action $a_{\text{branch}}$ based on the query, current path, and retrieved nodes $R(q)$. Otherwise, the action defaults to CONTINUE.

$$a_{\text{branch}} = \begin{cases} \Phi(q_{t+1}, \text{Path}(n_{\text{cur}}), R(q_{t+1})) & \text{if } H_{\text{filter}} \text{ is true} \\ \text{CONTINUE} & \text{otherwise} \end{cases}$$

The possible actions are:

– **CONTINUE**: Add a new node to the current branch.
– **CREATE_BRANCH**: Start a new branch from the fork point $n^*_{fork}$.
– **SWITCH_BRANCH**: Switch the active branch to the one containing $n^*_{fork}$.

• **Step4: Context Construction** The final context $C_{t+1}$ is constructed by combining the full dialogue of the current active path with summaries of inactive branches and topics. This provides focused, relevant information while maintaining a broad overview of the entire conversation. The context is formed as:

$$C_{t+1} = \text{Concat}\big(\{c_i \mid n_i \in \text{Path}(n_{\text{cur}}, T_{\text{act}})\}\big) \bigoplus_{\substack{B_j \in T_{\text{act}}, \\ B_j \neq B_{\text{act}}}} S(B_j) \bigoplus_{\substack{T_k \in H_t, \\ T_k \neq T_{\text{act}}}} S(T_k)$$

This structured context includes: (1) The complete dialogue history of the current active path. (2) Summaries of all other branches within the active topic tree. (3) Summaries of all other topic trees in the conversation history.

## 4 NON-LINEAR TASK MULTITURN DIALOGUE (NTM) BENCHMARK

While existing multi-turn dialogue datasets have been instrumental, they are often inadequate as they typically feature a limited number of turns with an average length of under 10 turns and assume a fixed, linear context (Deshpande et al., 2025; Kwan et al., 2024; Bai et al., 2024). This fails to capture the complexity of long-running conversations with dynamic topic shifts, making them unsuitable for assessing a model's robustness and long-range contextual reasoning. To bridge this gap, we introduce the Non-linear Task Multiturn Dialogue (NTM), a benchmark designed to test the limits of contextual understanding and robustness in LLMs.

### 4.1 DATA CREATION

NTM comprises a collection of dialogues focused on two domains: daily life planning and coding support. The dataset was constructed using state-of-the-art LLMs leveraging few-shot prompting to generate initial dialogues. Subsequently, each dialogue underwent a rigorous process of manual review, polishing, and filtering by human annotators to ensure high quality and task complexity.

Crucially, NTM dialogues focus on two significant aspects: Topic shifts and Instruction Refinement, which are common in real-world conversations but often overlooked in existing datasets.

• **Topic Shifts**: Each dialogue is designed to include multiple topic shifts. These shifts are not random but are contextually relevant, reflecting how real conversations evolve. For example, a dialogue may start with planning a trip and then shift to discussing dietary preferences for the trip.

• **Instruction Refinement**: The dialogues also incorporate instances where users refine or change their instructions based on previous responses. This aspect tests the model's ability to adapt to evolving user needs and maintain coherence throughout the conversation.

This design ensures that NTM evaluates not just information recall, but a model's ability to maintain focus and adapt to a dynamically evolving conversational landscape.

### 4.2 KEY CHARACTERISTICS

NTM is distinguished by the following features:

• **Extended Dialogue Length**: The dataset includes a total of 165 dialogues with about 3000 turns, covering 10, 15, 20, and 25 rounds of conversations, which provide a clear measure of model scalability as context grows.

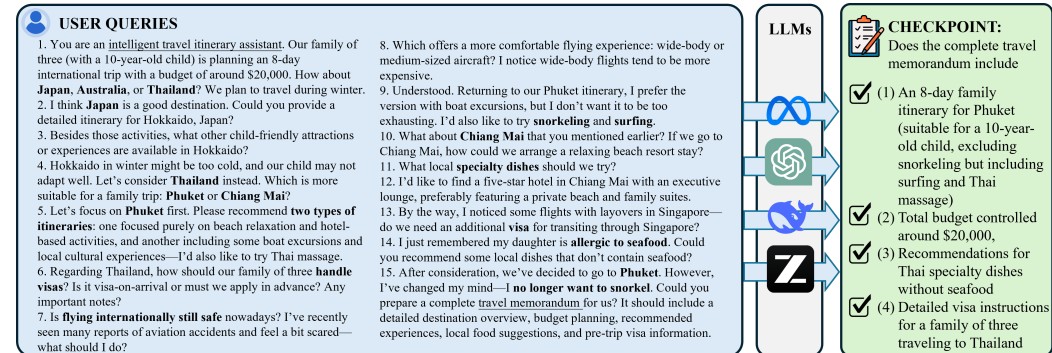

Figure 3: An example of a 15-turn dialogue from the NTM, illustrating topic shifts and instruction refinements. It is about planning a family trip, where the user changes their preferences, adds new requirements and shifts topics as the dialogues progresses. On the right side is the corresponding checkpoint questions for evaluating task completion, which are designed to objectively assess whether the model has fulfilled the user's final request. More details are in Appendix B.4.

- **Topic Dynamics**: Each dialogue contains multiple topic shifts and instruction refinements, challenging models to maintain coherence and relevance in a non-linear conversational flow.

- **Task-Oriented Focus**: Every dialogue culminates in a clear task that requires accurate information synthesis from the preceding conversation, enabling objective evaluation through task completion metrics.

### 4.3 EVALUATION METRICS

We evaluate the performance from 2 perspectives: task completion accuracy and token efficiency.

- **Task Completion Rate (TCR)**: Our primary metric for task success. Each task in the NTM benchmark is decomposed into at least three verifiable checkpoints(a yes/no decision). TCR is the average completion rate across these checkpoints, providing a robust measure of task fulfillment. This annotated metric provides a more robust and interpretable measure of a model's true task-fulfillment capabilities compared to relying solely on scores from a judge LLM.

- **Average Context Tokens (ACT)**: Measures the average number of context tokens used per turn. It quantifies context efficiency, with lower values indicating better performance, which is crucial for managing long dialogues under token and cost constraints.

### 4.4 COMPARISON WITH EXISTING DATASETS

To highlight the unique features of NTM, we compare it with existing multi-turn dialogue datasets in Table 1. NTM stands out with its significantly longer average and maximum turn counts, as well as its focus on non-linear dialogue evolution, which is absent in other datasets. This makes NTM a more challenging and realistic benchmark for evaluating the capabilities of dialogue systems in handling complex, multi-turn interactions.

| Dataset | Avg. Turns | Max Turns | Total turns | Non-linear Evolution |
|---------|------------|-----------|-------------|----------------------|
| Multichallenge | 5 | 10 | 1365 | No |
| MT-Eval | 7 | 14 | 1170 | No |
| MT-Bench-101 | 3 | 7 | 4208 | No |
| **NTM (Ours)** | **18** | **27** | **2929** | **Yes** |

Table 1: Comparison of NTM with existing multi-turn dialogue datasets.

## 5    EXPERIMENTAL SETUP

To rigorously evaluate the performance and efficiency of our proposed Context-Agent framework, we designed a comprehensive set of experiments. The primary goal is to demonstrate our model's superior ability to manage context in the long-form, non-linear dialogues that current benchmarks fail to represent.

The experiments were designed to answer the following research questions: (1) How does Context-Agent perform on complex, long-horizon dialogue tasks compared to baseline methods? (2) To what extent can Context-Agent improve token efficiency without compromising task success? (3) What are the individual contributions of the tree-based structural representation and the RAG-based retrieval mechanism to the overall performance of our system?

### 5.1    EVALUATION BENCHMARK

A significant challenge in evaluating long-turn conversational models is the lack of suitable benchmarks. Existing datasets typically feature short, linear dialogues that do not adequately test a model's ability to handle complex, evolving conversations. And the most important reason is that their context offered to the model is usually a fixed-length linear sequence, which cannot reflect the advantages of our Context-Agent in managing non-linear dialogue history. Therefore, all models are evaluated on our newly proposed Non-linear Task Multi-turn Eval (NTM) benchmark.

### 5.2    BASELINE METHODS

We benchmark our Context-Agent framework against mainstream context management methods nowadays, which can be categorized into three groups:

- **Full History Concatenation (Full-History)**: This method involves concatenating the entire dialogue history as input to the model. While it provides complete context, it is computationally expensive and often impractical for long conversations due to token limits.
- **Truncation (Truncation)**: This approach retains only the most recent $k$ turns of the conversation, discarding earlier context. It is efficient but risks losing important information from previous dialogue turns. In our experiments, we set $k = 4$.

To ensure a comprehensive evaluation of our Context-Agent across different models, we conducted experiments on four recent and diverse LLMs: `GPT-4.1` (OpenAI, 2025), `DeepSeek-V3` (Liu et al., 2024a), `GLM-4-Plus` (GLM et al., 2024), and `Llama 3.1-70B` (Grattafiori et al., 2024). This selection includes both open- and closed-source models with varying context window sizes. For fairness and efficiency, all evaluations were performed with reasoning-disabled settings.

| Model | Open Source | Context Window |
|-------|:-----------:|:--------------:|
| GPT-4.1 | × | 1000k |
| DeepSeek-V3 | ✓ | 64k |
| GLM-4-Plus | ✓ | 128k |
| Llama 3.1-70B | ✓ | 128k |

Table 2: Details of the LLMs used

### 5.3    IMPLEMENTATION DETAILS

**Prompt Format:** All models receive the same system prompt instructing them. No chain-of-thought or explicit instruction tuning is applied to ensure fair comparison. More details are in Appendix B.3.

**Local Models:** To balance processing efficiency and accuracy, the Context-Agent's internal modules utilize lightweight local models. Specifically, we employ `gemma3-12B` (Team et al., 2025) for decision-making and `gemma3-4B` for summary generation. For dialogue context encoding, we use `Qwen3-Embedding-0.6B` (Yang et al., 2025), a lightweight, high-performance embedding model. Based on empirical tuning with these models, the similarity threshold $\theta_{\text{sim}}$ was set to 0.6. All experiments were conducted with an NVIDIA A100 40GB GPU.

**Evaluation Protocol:** To ensure both scalability and human-aligned judgment, we adopt a triangulated evaluation protocol combining human annotators and two state-of-the-art Judge LLMs: `GPT-5` (OpenAI, 2025) and `Gemini-2.5-Pro` (Comanici et al., 2025). We compute Cohen's $\kappa$ (Cohen,

1960) between Judge LLM and human labels. The result shows that the Cohen's $\kappa$ is as high as 0.96, indicating strong agreement and validating the reliability of our evaluation approach.

# 6 RESULTS AND ANALYSIS

## 6.1 MAIN RESULTS

The main results of our experiments are summarized in Table 3. Across all 4 LLMs, our Context-Agent consistently outperforms the Truncation method by a significant margin in terms of Task Completion Rate (TCR). It shows that except for the case of GPT-4.1 has a slight decrease of 3.3%, our method not only recovers the performance loss caused by truncation but also surpasses the Full-History method, achieving relative TCR improvements of 6.4%, 7.2%, and 7.0% on DeepSeek-V3, GLM-4-Plus, and Llama 3.1-70B, respectively. The reason why our method does not outperform Full-History on GPT-4.1 may be that GPT-4.1 has a very large context window (up to 1 million tokens), making it less sensitive to context management strategies. But even in this case, our method only incurs a minor 3.3% performance drop while significantly reducing token usage, demonstrating its robustness and efficiency. Especially for super long context windows LLMs like GPT-4.1, maintaining a full history is often impractical due to computational costs and latency. And our Context-Agent provides a alternative that balances performance and efficiency.

| Model | Method | TCR (%) ⇑ | TCR Gain(%) | ACT ⇓ | | | | ACT Drop (%) |
|---|---|---|---|---|---|---|---|---|
| | | | | 10-turn | 15-turn | 20-turn | 25-turn | |
| GPT-4.1 | Full-History | **87.4** | – | 4058 | 6341 | 9715 | 12673 | – |
| | Truncation | 61.2 | -30.0 | 1919 | 2252 | 2841 | 3376 | – |
| | **Context-Agent** | 84.5 | **-3.3** | 2108 | 3004 | 4157 | 6312 | **-52.0** |
| DeepSeek-V3 | Full-History | 62.3 | – | 3559 | 5378 | 7680 | 10515 | – |
| | Truncation | 43.6 | -30.0 | 1732 | 2096 | 2655 | 2978 | – |
| | **Context-Agent** | **66.3** | **+6.4** | 1935 | 2917 | 4314 | 6973 | **-42.2** |
| GLM-4-Plus | Full-History | 72.7 | – | 4512 | 6896 | 9253 | 11873 | – |
| | Truncation | 48.1 | -33.8 | 2733 | 3438 | 3819 | 4570 | – |
| | **Context-Agent** | **77.9** | **+7.2** | 1922 | 3058 | 4752 | 7638 | **-49.3** |
| Llama 3.1-70B | Full-History | 67.1 | – | 3496 | 5088 | 7170 | 8920 | – |
| | Truncation | 47.9 | -28.6 | 1628 | 1975 | 2445 | 2883 | – |
| | **Context-Agent** | **71.8** | **+7.0** | 2187 | 2638 | 3975 | 4784 | **-44.1** |

Table 3: Main results on the NTM benchmark across different LLMs and context management methods. TCR Gain shows the relative percentage change compared to Full-History. ACT columns show Average Context Tokens for 10/15/20/25-turn dialogues; ACT Drop is the average percentage reduction of Context-Agent compared to Full-History.

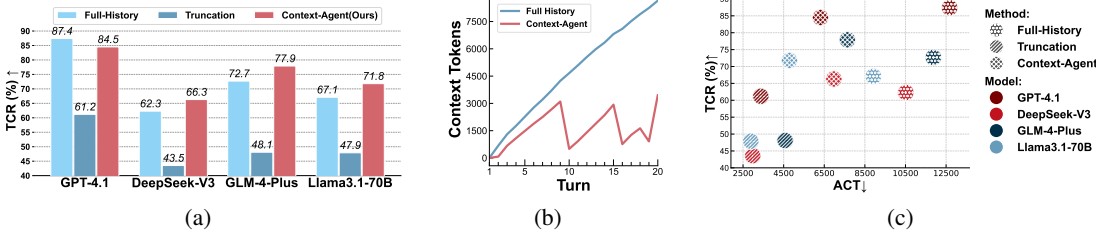

Figure 4: (a) TCR comparison across different methods and models. (b) A typical example of context tokens change trend in a 20-turn dialogue. (c) Trade-off between TCR and ACT, where the ideal point is the top-left corner (high TCR, low ACT).

Another notable observation is that though another 3 open-source models (DeepSeek-V3, GLM-4-Plus, and Llama 3.1-70B) still have considerable context windows (64k or 128k tokens), and the total context length of our NTM benchmark is lower than these limits, their TCR

scores with Full-History are still significantly lower than that of `GPT-4.1`. This indicates that merely having a large context window does not guarantee effective utilization of context, especially in complex, non-linear dialogues. Our Context-Agent has demonstrated its ability to effectively manage and utilize context, leading to substantial performance gains.

From these results, we can draw several key insights:

- **Effectiveness of Context-Agent**: The consistent TCR improvements across different models and dialogue lengths demonstrate that Context-Agent effectively manages context in complex, long-horizon dialogues. It not only recovers the performance lost due to truncation but also surpasses the full-history approach in most cases.

- **Token Efficiency**: The significant reductions in ACT indicate that Context-Agent is highly efficient in utilizing context. By intelligently selecting relevant information through its tree structure and RAG mechanism, it minimizes unnecessary token usage while still providing sufficient context for accurate responses.

- **Robustness Across Models**: The performance gains observed across a diverse set of LLMs, including both open-source and closed-source models with varying context window sizes, highlight the robustness and generalizability of the Context-Agent framework.

## 6.2 ABLATION STUDIES

To understand the source of Context-Agent's effectiveness, we conducted an ablation study.

The first one is apply RAG on Dialogue History, but remove the tree structure, forcing the RAG to retrieve from a linear sequence of turns (w/o Tree). We dynamically set $k \in \{3, 5\}$ based on dialogue depth: $k = 3$ for 10-15 turns, $k = 5$ for 20-25 turns, to balance recall and noise suppression. The second ablation removes the RAG retriever and relies solely on a heuristic for branch decision (w/o RAG). This tests whether the tree structure alone can provide sufficient context selection.

Table 4 summarizes the results of the ablation study on `DeepSeek-V3`. Both ablations lead to significant drops in TCR compared to the full Context-Agent model. Removing the tree structure results in a 35.0% decrease in TCR. This highlights the importance of the tree structure in organizing the interal logical flow of the dialogue, enabling effective context selection. Without it, the RAG can only get the result that may only has semantic relevance but lacks logical relevance.

| Method | TCR (%) | TCR Drop (%) |
|---|---|---|
| Full-History | 62.3 | - |
| w/o Tree | 40.5 | -35.0 |
| w/o RAG | 45.3 | -27.3 |
| Context-Agent | 66.3 | +6.4 |

Table 4: Ablation study results on Deepseek-V3

Removing the RAG retriever leads to a 27.3% drop in TCR, indicating that the heuristic alone is insufficient for accurate branch decision-making. The RAG mechanism provides critical context by retrieving semantically relevant historical nodes, and at the same time, with the heuristic filtering, it largely avoids misjudgments when selecting the fork point.

## 7 CONCLUSION

In this paper, we addressed the critical limitation of conventional linear context management in handling the non-linear flow of multi-turn dialogues. We introduced Context-Agent, a novel framework that represents dialogue history as a dynamic tree structure, augmented by a retrieval mechanism. This approach successfully models the hierarchical and branching nature of human conversations, enabling effective navigation of complex interactions involving topic shifts and refinements. Our extensive experiments on the newly proposed NTM benchmark demonstrate that Context-Agent consistently outperforms traditional context management methods across various LLMs, achieving significant improvements in task completion rates while drastically reducing token usage. Ablation studies confirm the critical contributions of both the tree structure and RAG components to the overall performance. Our work underscores the potential of structured context management and offers a promising direction for developing more robust and efficient dialogue systems capable of handling long-horizon, dynamic conversations.

# REPRODUCIBILITY STATEMENT

All experimental projects in this paper are reproducible. We will release the code, data, and prompts used in our experiments to facilitate future research. The specific details of the models, datasets, and evaluation protocols are thoroughly documented in the paper and its appendices. We also provide detailed instructions for setting up the experimental environment and running the code to ensure that other researchers can replicate our results. The relevant materials is available at `https://anonymous.4open.science/r/Context-Agent-and-NTM-Benchmark-01C4/`.

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

# A  LLM USAGE STATEMENT

In the preparation of this paper, we utilized Large Language Models (LLMs) as assistive tools to mainly enhance research efficiency. We explicitly detail all uses of LLMs below and affirm that all core intellectual contributions, including the formulation of hypotheses, experimental design, analysis of results, and the final conclusions, were conceived and executed by the human authors.

The specific applications of LLMs in our work are as follows:

**Literature Review and Scoping**: In the initial research phase, we employed LLMs (e.g., Google's Gemini Deep Research) to conduct preliminary explorations of the research landscape surrounding "modeling non-linear conversational flows in multi-turn dialogues". This involved identifying seminal papers and outlining the evolution of relevant techniques. This process accelerated our acquisition of a broad understanding of the field. All cited literature was subsequently verified, read, and critically analyzed by the authors.

**Data Analysis and Script Generation**: We utilized LLMs to assist in writing and debugging scripts for data preprocessing and experimental analysis. Specific tasks included parsing log files to extract performance metrics. All LLM-generated code was thoroughly reviewed, tested, and modified by the authors to ensure its correctness, efficiency, and alignment with our experimental setup.

The primary LLMs used in this research include: OpenAI's `GPT-5` and Google's `Gemini-2.5-Pro`.

# B APPENDIX

## B.1 CONTEXT-AGNET LATENCY AND TRADE-OFF ANALYSIS

Beyond token efficiency, we analyzed the end-to-end response latency to provide a complete picture of Context-Agent's practical performance. Our method's hybrid architecture involves several calls to local, lightweight language models for tasks such as branch decision-making and node summarization, which introduces time overhead compared to the baseline's single API call.

However, the latency of the full-context baseline is not constant; it degrades as the dialogue history grows and the token payload for the API call increases. This degradation partially offsets the inherent overhead of our method. To quantify this trade-off, we measured the average response time on a single NVIDIA A100 40GB GPU for the 20-turn dialogue scenario. The following table summarizes the average response times:

| Method | Average Response Time (s) | Relative Increase (%) |
|---|---|---|
| Full-History | 12.5 | - |
| Context-Agent | 13.5 | +8.0% |

Table 5: Average response time for different context management methods on a 20-turn dialogue.

Our experiments indicate that Context-Agent incurs a modest 8% increase in average response time. We argue this represents a highly favorable trade-off, given the substantial improvements in token efficiency. It is important to note that these measurements were conducted on a single A100 40GB GPU. This latency overhead could likely be mitigated in a production environment through optimizations such as deploying on enterprise-grade hardware or utilizing lightweight models fine-tuned for the specific decision and summarization sub-tasks.

## B.2 THE DETAILED ALGORITHM OF CONTEXT-AGENT

The complete algorithm of the Context-Agent framework is presented in Algorithm 1. It outlines the step-by-step process of managing dialogue context, including topic and branch management, node updates, and context construction.

---

**Algorithm 1** Context-Agent Framework

---

**Require:** Dialogue history $H_t$, User query $q_{t+1}$
**Ensure:** Constructed context $C_{t+1}$
    **1. Topic and Branch Management**
1: $(a_{\text{topic}}, T_{\text{target}}) \leftarrow \Psi(q_{t+1}, \{S(T_i)\}_{T_i \in H_t})$         ▷ Topic decision
2: Update $T_{\text{act}}, n_{\text{cur}}$ based on $a_{\text{topic}}$
3: $n^*_{fork} \leftarrow \arg\max_{n_i \in T_{\text{act}}} \text{Sim}(\epsilon(q_{t+1}), v_i)$         ▷ Find fork point
4: **if** $H_{\text{filter}}(n^*_{fork}, n_{\text{cur}})$ **then**
5:     $a_{\text{branch}} \leftarrow \Phi(q_{t+1}, \text{Path}(n_{\text{cur}}), R(q_{t+1}))$         ▷ Branch decision
6: **else**
7:     $a_{\text{branch}} \leftarrow \text{CONTINUE}$
8: **end if**
9: Update $B_{\text{act}}, n_{\text{cur}}$ based on $a_{\text{branch}}$ and $n^*_{fork}$
    **2. Node Update**
10: Create new node $n_{\text{new}}$ as child of $n_{\text{cur}}$
11: $s_{\text{new}} \leftarrow S_{\text{node}}(n_{\text{new}})$         ▷ Summarize new node
12: $n_{\text{cur}} \leftarrow n_{\text{new}}$
    **3. Context Construction**
13: $C_{\text{path}} \leftarrow \{c_i \mid n_i \in \text{Path}(n_{\text{cur}})\}$         ▷ Content of active path
14: $C_{\text{inactive}} \leftarrow \{S(B_j) \mid B_j \neq B_{\text{act}}\} \cup \{S(T_k) \mid T_k \neq T_{\text{act}}\}$   ▷ Summaries of inactive parts
15: $C_{t+1} \leftarrow \text{Concat}(C_{\text{path}}, C_{\text{inactive}})$
16: **return** $C_{t+1}$

---

## B.3 Model Implementation Details

This section provides the specific prompts used to guide the lightweight language models for decision-making and summarization within the Context-Agent framework.

**Prompt for Topic Decision** The following prompt is used to instruct the topic decision model $\Psi$ to analyze the user's query against the summaries of existing topic trees. The model must determine whether the query initiates a new topic, continues the current one, or switches to a previous one.

```
# STRICT INSTRUCTION - EXECUTE ONLY THE FOLLOWING LOGIC CHAIN
Act as a dialogue topic consistency adjudicator. Your task is to objectively score the
semantic relationship between a new query from user and conversation history summary of
dialogues between user and AI assistant. You MUST perform exactly three steps:
1. [Theme Check] Does the new query discuss the SAME physical/conceptual core object as
history?
Valid: "battery life" → "charging speed" (core object = battery)
Invalid: "Beijing weather" → "Shanghai weather" (core object changed)
Rule: Disregard surface differences (tools/locations/times).
e.g., "Python data cleaning" vs "Excel data cleaning" → Invalid

2. [Continuity Check] Does the new query depend on historical context?
Valid: "How fast does it charge?" (refers to prior "battery")
Invalid: "Recommend restaurants" (no contextual link)
Rule: Specially verify pronouns (it/this/that/them etc.), probing words (how/why), some
specific signpost words (such as "return to", "previously mentioned", etc.), logical
progression

3. [Final Judgment] Output "yes" ONLY if both steps pass, otherwise "no"

# ANTI-ERROR PROTOCOLS (Critical for lightweight LLMs)
ABSOLUTELY PROHIBITED:
• No keyword matching (e.g., "weather" in different cities)
• No intent speculation (textual content only)

Core Object Definition (Key innovation):
- Physical: Devices/items/body parts (iPhone battery, car engine)
- Conceptual: Problems/tasks/themes (data cleaning, travel planning)
- Critical: Core object changes when tools/locations shift

# EXTENDED EXAMPLE BANK
| History Summary | New Query | Theme | Continuity | Output |
|----------------------|-----------------|-------|--------|--------|
| "iPhone 15 battery life" | "Charging speed?" | Yes | Yes | yes |
| "Beijing weather today" | "Shanghai temperature?" | No | No | no |
| "Python Pandas cleaning" | "Excel missing values" | No | No | no |
| "Avatar movie effects" | "Cameron's next film?" | Yes | Yes | yes |
| "Diabetes diet tips" | "Exercise recommendations"| Yes | Yes | yes |
| "Laptop overheating" | "Phone thermal issues" | No | No | no |

# OUTPUT REQUIREMENTS
Only output SINGLE word: yes or no WITHOUT any extra characters (no spaces/punctuation)

# CURRENT INPUT
Now, please start comparing the history summary and the new query:
History Summary: {summary}
New Query: {query}
```

**Prompt for Branch Decision**    The branch decision model $\Phi$ is prompted to evaluate the user's query in the context of the current dialogue path and the most relevant historical nodes. The model must decide whether to continue the current branch, create a new branch, or switch to an existing one.

```
# Role and Task
You are a Dialogue Flow Controller. Your core task is to analyze the user's query and
conversational context to determine their navigational intent. You MUST output ONLY a
single JSON object as your decision, with no additional content.
# Core Decision Rules
Decisions must be based on comparing "Retrieved History Nodes" with the "Current Path",
following these specific rules:
1. If the user's query is most relevant to a "historical node" (retrieved ancestor node),
shows a tendency to diverge from the "Current Path", and the content of the current path
provides no substantial help in answering the new query (i.e., the presence or absence
of current path content makes no significant difference to the answer) → MUST create a
new branch.
2. If the user's new query is highly similar to a historical node in a non-current topic
branch, or if the user explicitly expresses a desire to return to a existing topic
branch, and providing the context of the previously existing topic branch is obviously
helpful for answering this new query. → MUST switch to the branch that the retrieved
history node belongs to.
3. If the user's query is a logical continuation of the "last turn in the Current Path"
and the current path context helps better answer the new query → continue along the
current path.
# Input Information
## Existing Branches
{existing_branches}
## Current Path Summaries
{current_path_json}
## Retrieved History Nodes
{rag_results_json}
## New User Query
"{user_query}"
# Output Requirements
Choose one of the following actions and output your decision as a single JSON object
with these fields:
- CONTINUE: User continues the current topic. Use ONLY when the query is a direct,
incremental continuation of the "last turn in the Current Path".
→ JSON structure: {{"action": "CONTINUE", "reason": "[Explanation for continuing]"}}
- CREATE_BRANCH: User wants to diverge from a past decision point. Must provide the fork
node ID (fork_node_id). Use when the user clearly "backtracks" or "pivots" to explore an
alternative path from an earlier conversation node (default choice).
→ JSON structure: {{"action": "CREATE_BRANCH", "fork_node_id": "[ID of most relevant
historical node]", "reason": "[Explanation for creating new branch]"}}
- SWITCH_BRANCH: User wants to switch to another existing branch and providing the
context of the previously existing topic branch is obviously helpful for answering this
new query. Must provide the target branch ID that the retrieved history node belongs to .
→ JSON structure: {{"action": "SWITCH_BRANCH", "target_branch_id": "[Target branch ID]",
"reason": "[Explanation for switching]"}}
# Example References
## Example 1: Create New Branch
Query: "I think Beijing is too cold. Let's check out Guangzhou instead."
Decision:
{{"action": "CREATE_BRANCH",
"fork_node_id": "{most_relevant_rag_node_id}",
"reason": "User rejects the current path ('too cold') and pivots to an alternative
('Guangzhou') from the retrieved node '{most_relevant_rag_node_id}'. Additionally,
previous discussions about Beijing travel plans provide no help in formulating a new
plan for Guangzhou."}}
## Example 2: Continue Current Branch
Query: "Okay, besides the Palace Museum, what other historical sites do you recommend in
Beijing?"
Decision:
{{"action": "CONTINUE",
"reason": "The query is a direct continuation of the current topic (Beijing
attractions)."}}
## Example 3: Switch to Existing Branch
Query: "Let's pause on Beijing for now and go back to the Shanghai plan we discussed."
Decision:
{{"action": "SWITCH_BRANCH",
"target_branch_id": "branch_xyz",
"reason": "User explicitly requests to resume another existing branch ('Shanghai plan').
The previous discussion about Shanghai is obviously helpful for answering the new
query."}}
Please output the final decision as a JSON object based on the provided context.
```

**Prompt for Node Summarization**     The node summarization model $S_{node}$ is prompted to generate concise summaries of dialogue nodes. The prompt emphasizes the need for brevity and relevance, ensuring that the summaries capture the essence of each node for effective context management.

```
# Role and Goal
You are a conversation summary generator. Your sole task is to compress a single
conversation between [USER] and [ASSISTANT] into the following format. This object will
be used by another language model to quickly understand the context of past
interactions.You only need to output the content of summary_for_llm, with no other
content whatsoever.
# Output Format
summary_for_llm: (string) An information-concentrated summary designed for AI, written
in the third person. This summary must integrate the user's intention and the LLM's
actions into a single declarative sentence. This is the most critical field for
providing context to downstream LLMs. If specific numbers are involved, the summary must
include important data.
# Example
## Input Conversation:
[USER]:
I'm trying to understand the difference between 'git merge' and 'git rebase'. Can you
explain it to me in simple terms, maybe with an analogy?

[ASSISTANT]:
Certainly. You can think of a project's history as a storybook.
- **`git merge`** is like connecting two separate story chapters (branches) with a new
page that says, "At this point, the stories from Chapter A and Chapter B merge." It
creates a new "merge commit" and preserves the exact history of both branches, resulting
in a history graph that looks like a diamond or a fork.
- **`git rebase`** is like taking all the pages of your Chapter B and placing them one
by one at the end of Chapter A. It rewrites the history of your branch to make it look
as if all your work was done in a straight line. This creates a cleaner, linear history.
In short, `merge` preserves history, while `rebase` rewrites history for the sake of
linearity. For team collaboration, `merge` is usually safer.

## Expected Output:
summary_for_llm: "The user asked about the difference between 'git merge' and 'git
rebase', and the AI assistant clarified their respective functions and different impacts
on commit history by providing definitions and analogies."

# Task
Now, please analyze the following conversation:
[USER]:
{user_message}
[ASSISTANT]:
{assistant_message}
```

### B.4 NTM BENCHMARK DETAILS

The Non-linear Task Multiturn Dialogue (NTM) benchmark is designed to evaluate the performance of dialogue systems in handling complex, multi-turn conversations with dynamic topic shifts and instruction refinements. Below are the details of the NTM benchmark.

#### B.4.1 HUMAN ANNOTATION GUIDELINES

To ensure the quality and consistency of the NTM benchmark, human annotators reviewed, polished, and filtered the generated dialogues based on the following primary criteria:

- **Coherence and Naturalness**: The dialogue must flow logically and feel natural, avoiding robotic or repetitive responses. Topic shifts, a key feature of the benchmark, must be contextually plausible and not feel abrupt or random. The overall conversation should mimic the ebb and flow of genuine human interaction, including clarifications, refinements, and relevant digressions.

- **Task Complexity**: Each dialogue must build towards a clear, non-trivial final task. Successfully completing this task should require the model to synthesize and integrate information scattered across multiple turns, including handling user refinements and instruction changes. Simple, single-turn information retrieval is insufficient; the task must test long-range reasoning and memory.

- **Clarity and Objectivity of Checkpoints**: To facilitate objective and reproducible evaluation, the final task must be decomposable into a set of clear, unambiguous, and verifiable checkpoints. Each checkpoint should correspond to a specific sub-goal of the user's final request and be answerable with a simple "yes" or "no", minimizing subjective judgment during evaluation.

#### B.4.2 THE DETAILED TOPIC TREES

In the previous Figure 3 in Section 4, we provided a dialogue example. To more intuitively demonstrate the formation of the dialogue tree, we have visualized the dialogue example shown in Figure 3 into a tree structure.

Showed in Figure 5, the dialogue starts with planning a family trip. In the first turn, the user introduces the plan and suggests several potential destinations, which sets a potential fork point for the future exploration of different destinations. Then the user and the assistant discuss the details of the Hokkaido itinerary, including child-friendly attractions. However, in turn 4, the user shifts the topic to Thailand due to concerns about the cold weather in Hokkaido. This shift is still within the topic of trip planning but introduces a new destination. And it is totally different from the previous discussing about Japan. The history of the first three turns is not so useful for the following discussion about Thailand.

Therefore, the Context-Agent creates a new topic tree for Thailand, starting a new branch from turn 4. The user then explores two potential locations in Thailand: Phuket and Chiang Mai, requesting different types of itineraries and activities. This introduces another fork point at turn 5, where the user asks for two distinct itinerary options for Phuket.

In turn 7, the user raises a concern about the safety of international flights, which is totally different from the previous topic of trip planning. This prompts the Context-Agent to create another new topic tree for flight safety, starting a new tree from turn 7. The user and assistant discuss various aspects of flying, including aircraft types and comfort.

Then in turn 9, the user returns to the Phuket itinerary, indicating a switch back to the previous topic tree about Thailand. The Context-Agent recognizes this and switches the active topic tree back to Thailand. The user continues to refine their preferences for the Phuket itinerary, expressing a desire for a more relaxing experience without snorkeling. Nevertheless, in turn 10, the user again shifts the focus to Chiang Mai, asking about arranging a beach resort stay there. This indicates another switch within the Thailand topic tree. And in turn 14, the user refines their food preferences due to a seafood allergy. Finally, in turn 15, the user makes a final decision to go to Phuket but changes their mind about snorkeling and requests a comprehensive travel memorandum that synthesizes all the discussed information, including destination overview, budget planning, recommended experiences, local food suggestions, and visa information.

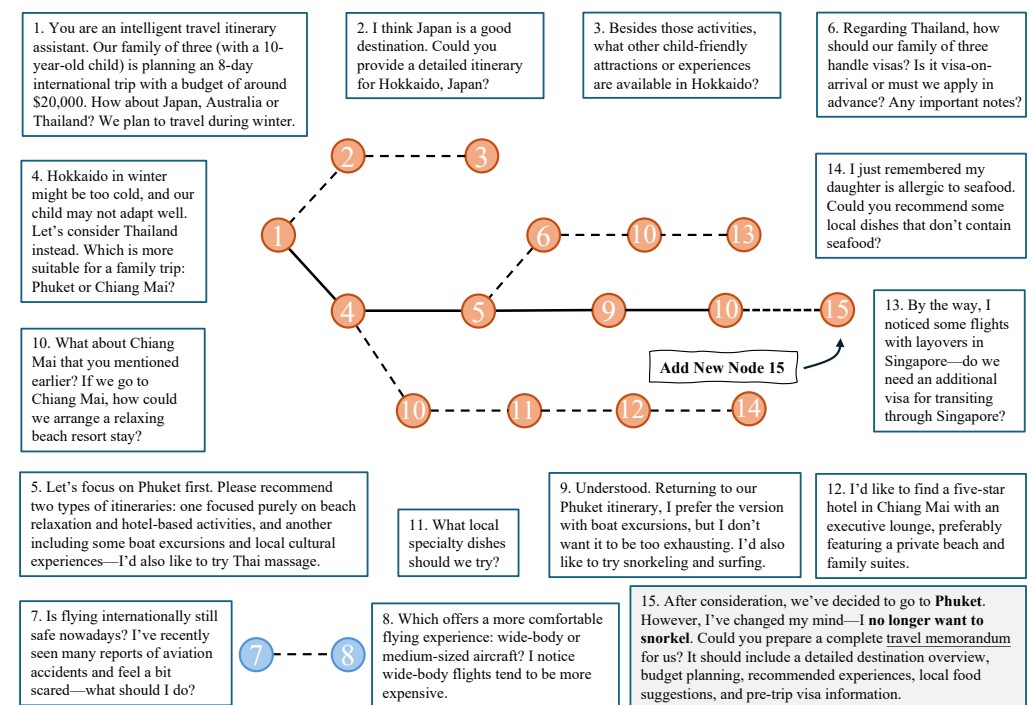

Figure 5: The topic tree structure corresponding to the dialogue example in Figure 3. Each node represents a turn in the dialogue, with branches indicating topic shifts and refinements. The solid edges represent the active path, while the dashed edges represent inactive branches.

### B.4.3 EXAMPLE FROM "CODING SUPPORT" DOMAIN

This example illustrates a typical dialogue from the NTM benchmark's coding support domain, featuring topic shifts and instruction refinements.

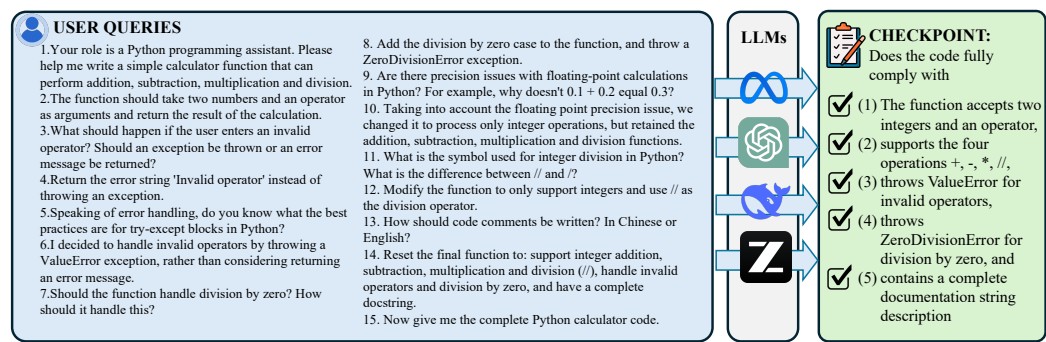

Figure 6: An example of a 15-turn dialogue from the NTM benchmark in the coding support domain. The dialogue features multiple topic shifts and instruction refinements, culminating in a clear task of generating a Python calculator function.

As shown in Figure 6, the dialogue begins with a request for a basic calculator. The user iteratively refines the requirements—adding error handling and changing data types from floats to integers—while also digressing to discuss 'try-except' best practices and commenting conventions. Finally, the user consolidates all refinements into a final request for the complete code. This example highlights the benchmark's focus on testing a model's ability to handle instruction changes, topic shifts, and integrate information from a non-linear dialogue history.

