# OpenReview forum: "Beyond Sequential Context: Navigating Non-linear Flow of Multi-turn Dialogues with Dynamic Context Tree"
_ICLR.cc/2026/Conference — ICLR 2026 Conference Withdrawn Submission_

### Official Review · Reviewer_wPbp · 2025-10-31

**Soundness:** 2
**Presentation:** 3
**Contribution:** 2
**Rating:** 6
**Confidence:** 3

**Summary:**

This paper introduces Context-Agent, a novel framework designed to address the challenges Large Language Models face in managing non-linear multi-turn dialogue flow. The core idea is to model dialogue history as a dynamic tree structure, augmented by a Retrieval-Augmented Generation (RAG) mechanism, which allows for more efficient and structured context selection. The authors also propose a new benchmark, the Non-linear Task Multi-turn Dialogue (NTM), to evaluate models in long-horizon, non-linear conversational settings.

**Strengths:**

- The proposed dynamic tree-structured context management method offers an intuitive approach that aligns well with how humans cognitively process complex, non-linear conversations, involving branching and revisiting topics.
- The design principle is also reflected in similar features found in advanced AI dialogue applications. This alignment enhances the method's interpretability and user-friendliness, indicating a strong potential to improve the overall user experience in dialogue systems.

**Weaknesses:**

1. The contribution of structured management in high-context LLMs is not convincing enough. The paper attributes the strong performance of GPT-4.1 (87.4% TCR with Full-History) to its "extremely large context window." This explanation, while plausible, inadvertently weakens the paper's core argument regarding the necessity of structured non-linear context management. It suggests that simply increasing context length might sufficiently mitigate the problem, rather than requiring the proposed architectural changes. The work does not sufficiently demonstrate the benefits of Context-Agent in scenarios where context length is not a bottleneck, or when the total token count of the non-linear dialogue is well within typical LLM context limits.
2. The baseline methods for comparison are limited. The authors only compare the proposed Context-Agent with Full-History and Truncation in Table 3, both of which are straightforward. More advanced methods should be included for comparison. Moreover, the paper's central premise is the difficulty LLMs face in handling the "non-linear flow" of multi-turn dialogues. While Full-History and Truncation are standard baselines, the latter primarily evaluates the impact of context length limitation due to information loss. The choice of Truncation as a key baseline, without further explanation, does not directly address how it specifically exacerbates the challenges of non-linear dialogue structures.

**Questions:**

1. Comprehensive Token Efficiency Assessment:
The paper uses Average Context Tokens (ACT) to measure token efficiency, which quantifies only the tokens sent to the main LLM. However, the Context-Agent framework employs several lightweight LLMs (Gemma 3-12B, Gemma 3-4B) for internal decision-making and summarization. These internal inference processes also incur token consumption and computational overhead. Could the authors clarify whether ACT includes these internal LLM-generated tokens? If not, could a quantitative analysis of this additional token overhead (e.g., as a percentage of total tokens) be provided to present a more comprehensive efficiency and cost assessment of the framework in a real-world deployment?
2. Do State-of-the-Art foundation models still face this problem of managing non-linear dialogue flow? The introduction posits that Large Language Models face a "fundamental challenge" in managing non-linear dialogue flow. Given that advanced flagship LLMs like GPT-5 and Gemini 2.5 Pro are now widely available, testing these models could further clarify whether this "fundamental challenge" persists for the most capable LLMs. This would strengthen the paper's motivation by more precisely defining the scope and urgency of the problem for state-of-the-art systems.
3. How about the performance in linear multi-turn dialogues? The paper focuses on addressing non-linear multi-turn dialogues but does not discuss the performance of Context-Agent in standard linear multi-turn dialogue scenarios. A robust and generalizable context management framework should ideally not incur performance penalties in simpler settings while solving complex problems. Could the authors provide an evaluation (e.g., through small-scale experiments or theoretical analysis) of Context-Agent's performance in purely linear multi-turn dialogues compared to baselines like Full-History? Significant overhead or performance degradation in linear settings could diminish the framework's overall value and applicability.

---

### Official Review · Reviewer_hpBs · 2025-10-31

**Soundness:** 2
**Presentation:** 3
**Contribution:** 2
**Rating:** 4
**Confidence:** 4

**Summary:**

This paper addresses the challenge of managing non-linear, multi-turn dialogues, where conversations branch, revisit old topics, or refine previous instructions. The authors argue that the standard linear-sequence context window is ill-suited for this, leading to inefficiency and coherence loss.

The paper proposes two main contributions:

1. Context-Agent: A novel framework that models the dialogue history as a dynamic "forest" of topic trees. It uses a hierarchy of lightweight models (for node summarization, topic decisions, and branch decisions) and a RAG mechanism to dynamically build and navigate this tree, constructing a relevant context for the main LLM at each turn.

2. NTM (Non-linear Task Multi-turn Dialogue) Benchmark: A new, manually-vetted benchmark of 165 long-form (up to 25 turns) task-oriented dialogues designed to specifically evaluate model performance in scenarios with topic shifts and instruction refinements.

**Strengths:**

1. The paper tackles a well-defined and significant limitation of current LLMs. Managing non-linear conversations is a key challenge for complex agents, and the linear context model is a known bottleneck.
2. NTM stresses non-linear topic shifts and instruction refinements, with objective checkpoint scoring (TCR) and a practical efficiency metric (ACT).
3. The ablation in Table 4 provides strong initial evidence for the framework's design, showing that both the tree structure ("w/o Tree") and the RAG-based branch decision ("w/o RAG") are critical to its success.

**Weaknesses:**

1. Baseline scope. Core baselines are Full-History and Truncation. It's not enough. should compare it with some other memory management work such as DH-RAG mentioned or MemTree. Also, MemTree is a highly related work that should be discussed in the related work section.

2. Insufficient Related Work: Regarding Methods: MemTree or other tree-structured memory work should be covered. For Benchmark, TIAGE: A Benchmark for Topic-Shift Aware Dialog Modeling is also a Topic Shift benchmark.

3. The paper notes a 3.3% drop in TCR for Context-Agent compared to Full-History on GPT-4.1. While justified by a 52% token saving, this is a very important finding. It suggests that for highly capable models with very large context windows, this structured approach may lose critical information, leading to worse task performance. This trade-off should be discussed more prominently as a limitation.

4. The tree structure incurs more memory overhead (storing both content and summaries) and multiple LLM calls per turn (topic/branch decisions, summarization). The paper reports only +8% latency but provides no memory usage analysis or cost breakdown. The trade-off between 6-7% performance gains and increased computational costs remains unquantified.

**Questions:**

1. I downloaded the Paper-Code.md file attached. The md file looks good to me. While, the anonymized repo could show the structure of the repo but no content. They are all showing that: The requested file is not found.not sure if it's a website error or the repo expired. Please fix it.

2. the paper needs memory-related baselines as mentioned in Weakness 1&2.

3. report more metric such as latency or memory in main context since that's my first concern for a tree-based structured or a hierarchical framework.

---

### Official Review · Reviewer_uNFn · 2025-11-01

**Soundness:** 2
**Presentation:** 2
**Contribution:** 2
**Rating:** 2
**Confidence:** 3

**Summary:**

In this paper, the authors propose a new dataset and framework for effectively modeling non-linear dialogue flow. The authors point out that previous research, which treats dialogue as a linear sequence, fails to capture the inherently non-linear nature of human conversation. To address this issue, the authors introduce Context-Agent, a framework that represents dialogue history as a dynamic tree structure, enabling hierarchical and branching representations of conversations. To evaluate this approach, the authors present a new dataset, Non-linear Task Multi-turn Dialogue (NTM), which reflects non-linear conversational dynamics. Experimental results on the NTM dataset demonstrate the effectiveness of the Context-Agent framework.

**Strengths:**

1. The authors propose a task that models non-linear dialogue flow to better reflect the nature of human conversation.
2. The authors introduce Context-Agent, a framework that models dialogues hierarchically through a tree structure, enabling the representation of branching and non-linear conversational sequences.
3. To evaluate Context-Agent, the authors present a new dataset called NTM, which captures non-linear conversational dynamics more effectively than existing datasets.

**Weaknesses:**

1. The authors evaluate Context-Agent only on the NTM dataset, which NTM dataset created to better capture the non-linear dynamics of dialogue. While this motivation is reasonable, it would strengthen the paper to include complementary evaluations on existing dialogue benchmarks to verify the effectiveness and generalizability of Context-Agent beyond the proposed dataset.
2. The experiments are conducted exclusively on (quite) large-scale language models under relatively constrained settings. Including smaller or more diverse baseline models would provide stronger evidence for the framework's effectiveness. In particular, if similar improvements were observed in smaller models, it would offer meaningful insights into the scalability and applicability of Context-Agent.
3. The framework uses several fixed internal components (e.g., embedding, etc.), but there is no ablation study examines their impact. Since the overall performance of Context-Agent may depend on the quality of these components, such an analysis is necessary to demonstrate the robustness and generalizability of the framework across different module configurations.
4. The NTM dataset is relatively small, and each dialogue contains only a limited number of turns. Consequently, it may not fully capture the dynamic and highly non-linear nature of real human conversation. Moreover, it does not address more complex, long-term, or multi-session conversations that exhibit deeper hierarchies and richer branching structures. This limitation reduces the dataset's representativeness and the overall contribution of the work.

**Questions:**

1. In Section 2, rather than simply listing related work, it might be more effective to conclude the paragraph by explicitly highlighting how this paper differentiates itself from prior studies.
2. Why are the topics in the NTM benchmark limited to daily life planning and coding support? Is there a particular reason for selecting these two domains?
3. I have a slight concern that a graph-based structure might be more suitable than a tree structure for modeling the non-linear nature of dialogue. In real-world conversations, multiple topics can often be interconnected, and the current tree-based representation might have difficulty capturing such multi-linked relationships. As recent studies increasingly adopt graph-based approaches for dialogue modeling, could the authors clarify how the proposed tree based framework offers a meaningful contribution in this context?
4. Why do the authors compare Context-Agent only against RAG-based baselines? It would be more informative to include comparisons with other types of context management or dialogue modeling methods, such as memory-based, hierarchical, or retrieval-free approaches, to more comprehensively evaluate the effectiveness of the proposed framework.
6. Please see the Weaknesses for additional questions.

---

### Official Review · Reviewer_WRsu · 2025-11-12

**Soundness:** 2
**Presentation:** 3
**Contribution:** 2
**Rating:** 2
**Confidence:** 4

**Summary:**

The paper proposes Context-Agent, which represents the multi-turn dialogue history as a dynamic tree structure and constructs a more selective and concise context with this structure.  The paper also presents a new benchmark, NTM, to evaluate dialogues with non-linear topic flows. The data are LLM-generated and human-reviewed/edit to ensure long, non-linear flows with topic shifts and instruction refinements.

Results show that Context-Agent improves task completion and reduces tokens compared to common baselines, though it introduces more system complexity, latency overhead, and shows mixed performance on models with huge context windows.

**Strengths:**

1. The paper presents a novel method that models the dialogue history with dynamic topic tree structure with topic trees and branches, adopts a two-stage branch decision strategy, and applies structure-aware retrieval to construct a more concise and relevant context.

2. The paper provides a new benchmark, NTM, that targets long, non-linear flows with topic shifts and instruction refinements and also defines clear metrics (TCR for effectiveness and ACT for efficiency).

3. The paper is clear and well-structured, making it easy to follow and understand.

**Weaknesses:**

1. All experiments are conducted on the newly proposed NTM benchmark, with no results on other public datasets, limiting generalization claims and it remains uncertain whether the method would not degrade across other types of TOD tasks.

2. The branch gate relies on a threshold and a heuristic, which may be unrobust / hard to tune across domains.

3. The tree-based routing works well for task-oriented non-linear flows but lacks explicit mechanisms for noise filtering and for integrating fine-grained evidence across multiple prior branches, since inactive branches contribute only summaries and there is no "merge across branches" action, which may hinder performance when users digress or ask to connect disparate branches.

**Questions:**

1. The paper adapts the term RAG to describe internal, structure-aware dialogue history, which is not a direct and clear usage of the term. Better to explicitly define / rebrand it and unify the terminology in the method and result sections.

2. The ablation study shows large drops when removing the tree or RAG. Can you further discuss which sub-modules matter most (topic decision, fork identification, heuristic gate, lightweight LM) through finer-grained ablations?

3. How sensitive are results to the similarity threshold $\theta_{sim}$ and the choice of embedding model?

4. How does the system behave under noisy or off-topic turns that are semantically close to old nodes (risking false positives)?

5. On GPT-4.1, Task Completion Rate drops slightly vs. full history; is there any way to further improve the method on such models, for example, with a hybrid method that falls back to full-history when below a confidence threshold?

---

### Note · Authors · 2025-12-27

I have read and agree with the venue's withdrawal policy on behalf of myself and my co-authors.